# The Influence of the Oral Microbiome on Oral Cancer: A Literature Review and a New Approach

**DOI:** 10.3390/biom13050815

**Published:** 2023-05-11

**Authors:** Anna Smędra, Jarosław Berent

**Affiliations:** Department of Forensic Medicine, Medical University of Lodz, 91-304 Lodz, Poland; jaroslaw.berent@umed.lodz.pl

**Keywords:** microbiome, alcohol, acetaldehyde, auto-brewery syndrome, oral cancer, risk factors

## Abstract

In our recent article (Smędra et al.: Oral form of auto-brewery syndrome. J Forensic Leg Med. 2022; 87: 102333), we showed that alcohol production can occur in the oral cavity (oral auto-brewery syndrome) due to a disruption in the microbiota (dysbiosis). An intermediate step on the path leading to the formation of alcohol is acetaldehyde. Typically, acetic aldehyde is transformed into acetate particles inside the human body via acetaldehyde dehydrogenase. Unfortunately, acetaldehyde dehydrogenase activity is low in the oral cavity, and acetaldehyde remains there for a long time. Since acetaldehyde is a recognised risk factor for squamous cell carcinoma arising from the oral cavity, we decided to analyse the relationship linking the oral microbiome, alcohol, and oral cancer using the narrative review method, based on browsing articles in the PubMed database. In conclusion, enough evidence supports the speculation that oral alcohol metabolism must be assessed as an independent carcinogenic risk. We also hypothesise that dysbiosis and the production of acetaldehyde from non-alcoholic food and drinks should be treated as a new factor for the development of cancer.

## 1. Introduction

The number of global cancer cases and deaths increases every year. The International Agency on Research in Cancer (IARC) provided the corrected GLOBOCAN 2020 with new estimates of the world cancer burden, indicating that in 2020, the number had increased to 19.3 million cases and 10.0 million deaths from cancer. Analogous data provided by GBD 2019 indicate 23.6 million cases and 10.0 million deaths. According to GLOBOCAN, the world burden of cancer is anticipated to reach 28.4 million new cases in the year 2040, an increase of 47% in comparison with 2020, with a more significant growth among transitioning countries (from 64% to 95%) compared to transitioned countries (from 32% to 56%) as a result of changes in demographics [1,2].

In 2020, cancers of the head and neck (HNCs) were responsible for more than 930,000 cases (approximately 700,000 males vs. 230,000 females) and 465.000 deaths (around 354,000 males vs. 113,000 females) worldwide [1]. HNCs occur primarily in the mouth, throat, and larynx. Other possible locations include the nose, sinuses, and salivary glands. It should be emphasised that in 2008, GLOBOCAN provided data indicating significantly lower rates of HNCs. At that time, slightly over 500,000 new HNC cases were identified yearly, making it the world’s sixth most common type of cancer [3]. This means that the frequency of HNC diagnoses has increased, possibly due to both population ageing and demographic growth but also due to shifts in the dissemination and spread of the significant risk factors.

Oral cancer (OC) is the most common neoplasm in the head and neck region [1,4,5]. OCs are defined as cancers of the tongue, the floor of the oral cavity, the lining of the cheeks, the roof of the oral cavity, the gums, and the lips but do not include laryngeal and pharyngeal cancers. The majority of OCs comprise oral squamous cell carcinomas (OSCCs) [6]. As stated by Vigneswaran et al., OSCCs derived from squamous cells account for 90% of all cancers in these locations [7]. The most common site of an OSCC is the floor of the mouth or the lateral/ventral tongue surface [8,9], but it also touches the lips, gums, cheeks, and hard palate. Other cancers of the oral region include lymphomas (3.5%) [10], sarcomas, small salivary glands, odontogenic tumours, and malignant melanomas [11].

In 2018, global estimates amounted to 354.864 new OC diagnoses and 177,384 deaths (a mortality rate of about 50%) [9,12,13]. In 2020, GLOBOCAN evaluated 377,713 new patients with OC worldwide (264,211 men vs. 113,502 women, with an age-standardised rate of 6.0 vs. 2.3 and a cumulative risk of 0.68% vs. 0.28, respectively) and 177,757 deaths, with a mortality rate of slightly lower than 50% (125,022 men vs. 52,735 women, with an age-standardised rate of 2.8 vs. 1.0 and a cumulative risk of 0.32 vs. 0.12, respectively) [1,4]. The corresponding figures given by the GBD are 373,000 cases and 199,000 deaths globally in the group “lip and oral cavity cancer” in 2019 [14].

Over the last few decades, the prevalence of OC has increased in many states. In men, it is the sixth most frequent morbidity (after prostate, lung, colorectum, liver, and stomach cancer) and the eighth most common mortality (following lung, prostate, liver, stomach, colorectum, Kaposi sarcoma, and oesophagus cancer) globally [1,4]. Regarding location, most cases are identified in Southwest and Central Asia, Eastern Europe, and Australia/New Zealand. Many analyses have shown that head and neck carcinomas are among the cancers with the highest prevalence, accounting for high incidence and fatality rates, notably in the Middle East, Asia, and Africa [15,16].

In states with a low or middle human development index (HDI) value, OC is the third leading form of cancer incidence (after prostate and lung) and the sixth main cause of mortality from cancer (after lung, liver, stomach, oesophagus, and prostate cancer) among men (in India, OC is the most frequent and deadliest form of cancer in males) [1,17,18,19]. Such elevated morbidity and death rates are the result of ordinary, lifestyle-related risk factors in the region, for example, smoking, betel-nut chewing, the consumption of alcohol [20,21], and infection-related and genetic factors [22]. For countries with a strong or very strong HDI, OC ranks fourteenth in terms of both cancer incidence and cancer mortality in males and fifteenth and fourteenth in females, respectively [1]. According to Shivivappa et al., cancer of the oral cavity and throat is Europe’s eighth most ubiquitous neoplasm and the eleventh major cause of mortality from cancer [23]. OC patients are typically diagnosed in their sixth or seventh decade of life. However, more people receive an OC diagnosis under 45 [20,24]. As shown above, there is a sex disproportion, as OCs are especially common in men [1,4,6,25,26].

After comparing the data provided by GLOBOCAN 2020 and the GBD 2019, some disparities are visible; they are minor in countries with well-established cancer registries and healthcare administrative systems and more prominent in countries from which GLOBOCAN and the GBD could not obtain high-quality or accessible data. Differences in the amount of OC can also be connected by grouping these cancers into different categories.

The above data indicate that OC has a significant and growing impact on the population’s health and will be an increasing problem in the future.

Despite progress in oncology studies and therapy, OC still has a dire prognosis. The treatment tends to cause serious complications for sufferers, both physical and physiological. Usually, these cancers are diagnosed when they are already advanced, which is unfortunately associated with a severe outcome (mean 5-year survival <50%) [27,28]. According to Neville et al., the overall 5-year survival rate in OC is just 50–55% [29], regardless of therapeutic advances. Data calculated in the United States of America in 2016 showed that just 63% of patients with oral and pharyngeal cancer were expected to live five years following their diagnosis [30]. In advanced cases, the 5-year survival rate is only 20% [31]. The high mortality associated with these cancers is primarily attributed to the symptomless presence of the illness in its initial phase [9]. Primary therapy is a surgical intervention, whereas radiotherapy and chemoradiotherapy are complementary procedures intended to constrict the tumour before surgery. Non-surgical treatment is usually palliative. So far, no screening tool suggested in the literature has improved the diagnosis and prognosis or helped reduce the incidence of OC [32]. The dire prognosis of these cancers underscores the significance of preventive efforts. The identification of individual carcinogens and risk pools, as well as the early detection of preneoplastic conditions, are crucial in cancer prevention [33,34].

In our review, we decided to describe the connections linking the microbiome, alcohol, and oral cavity cancer, referring to the case we reported in 2022. That case concerned a man who was found to produce alcohol in his mouth due to candidiasis. As we emphasised at the end of that article, ethanol, or more specifically, its metabolite acetaldehyde, is a carcinogen that may be an aetiological factor of mouth and throat cancers, which is why we find this issue an interesting problem to consider [35].

## 2. Materials and Methods

Our review focuses on associations between the microbiome, alcohol, and oral cavity cancer. We used a narrative review method, choosing to search the PubMed database using the following keywords: microbiome AND oral cancer; microbiome AND oral cavity cancer; microbiome AND alcohol; microbiome AND head and neck cancer. The search was conducted from 10 to 17 March 2023. Our objective was to analyse studies published in the last ten years. After a preliminary analysis, those that concerned a different topic were excluded. We added older matching articles that were references for the ones we searched for in the first step and which we considered relevant for this analysis. After removing duplicates, we found 166 articles in the PubMed database that fit the assumed thematic scope.

## 3. Oral Cavity Histology

The structure of the mucous membrane of the oral cavity comprises two strata: (a) the keratinised or non-keratinised stratified squamous epithelium and (b) the lamina propria of the mucosa. The stratified keratinised squamous epithelium covers the lamina propria in areas exposed to mechanical stimuli related to chewing (the gums, hard palate, and the back of the tongue) and consists of four layers (basal, spinous, granular, and keratinising). The non-keratinised stratified squamous epithelium covers the inner surfaces of the lips, cheeks, and sublingual area and consists of three layers (basal, spinous, and superficial). The lamina propria of the mucosa is made of fibrous connective tissue and contains small salivary glands (labial, buccal, palatal, and lingual). The proper mucosa contains clusters of lymphoid tissue (tonsils and loose lymphocytes), which protect against harmful factors from air and food [36].

## 4. Oral Cancer Risk Factors

Drinking alcohol, smoking tobacco, and infection with the human papillomavirus (HPV) (there are about 14 high-risk types of HPV, including HPVs 16, 18, 31, 33, 35, 39, 45, 51, 52, 56, 58, 59, 66, and 68; two of these, HPV 16 and HPV 18, are responsible for most HPV-related cancers, and HPV 16 in particular is responsible for oropharyngeal carcinoma) are most commonly recognised as the most critical risk factors for OC [24,37,38,39,40,41,42,43,44,45,46]. All the above-mentioned factors are controllable: it is a matter of lifestyle choice in the case of the first two and being vaccinated in the case of the latter. It should be noted that the meta-analysis by Arif et al. implied a negative synergy between HPV infection and smoking tobacco as well as HPV infection and high alcohol consumption, increasing the risk of primary OC development in a tiered and cumulative effect analysis [47]. Other risk indicators include Epstein–Barr virus (EBV, for nasopharyngeal carcinoma) [48,49], *Candida albicans* [50,51], a diet low in vitamin A and carotenoids, inadequate dental hygiene, including the regularity of tooth brushing, and chronic mouth trauma [52].

Another possible risk factor is the use of mouthwash [53]. Research studies examining this last correlation have conflicting outcomes, with some confirming and others disputing it. Carr and Aslam-Pervez noted that there is currently insufficient proof to reach any conclusive answers about the link between the use of an alcohol-containing mouthwash and the risk of OC [54]; however, Boffetta et al., for example, concluded that all mouthwash consumers had a mildly higher likelihood of developing this type of cancer [55]. In comparison, the study by Sorin et al. showed no statistically significant correlation between the use of mouthwash and the risk of OC. The only statistically significant correlation was between frequent use and OC [56]. A meta-analysis conducted in 2012 by Gandini et al. did not show a statistically significant relationship between the use of mouthwash and the risk of OC [57]. Aceves Argemi et al. recently performed an update on the meta-analysis and found no potential link between mouthwash use and an increased risk of OC [58].

However, among the many possible risks of HNC, alcohol and tobacco are the factors with the strongest associations [59,60]. Mello et al. state in their systematic review and meta-analysis that alcohol consumption and smoking tobacco are the primary risk factors for developing OC, with substantial synergistic effects [61]. This fact is not new and has been known for many years. For example, in the 1980s, Blot et al. remarked that traditional drivers of OC risk, such as drinking alcohol and smoking, significantly increase the risk of cancer when consumed simultaneously [62]. As previously noted, there are wide variations in the incidence of OC around the world; these variations are related to different lifestyles concerning the products used by the population, especially the types and amounts of alcohol consumed (different percentages of alcohol), as well as different styles of alcohol consumption (for example, daily consumption or occasional consumption) and the way that tobacco is used (in Europe and the USA, people usually smoke tobacco, while in Southeast Asia, people mostly use smokeless tobacco) [63].

The earliest evidence of alcoholic fermentation comes from 7000 BC [64]. As a matter of fact, over 60 various diseases and conditions have been causally associated with alcohol consumption [65]. In 2016, the World Health Organisation (WHO) evaluated that alcohol consumption was responsible for 3 million deaths worldwide [66]. It is worth noting that alcohol use exerts pleiotropic effects on human health which are highly dependent on the dose. Alcohol abuse can seriously affect mental [67,68,69] or physical health [70,71,72].

Even though alcohol is a notorious carcinogen, it remains underestimated by the population at large [73]. French pathologist Lamu published the first report on this relationship in Paris in 1910. It concerned a significant correlation between heavy drinking (absinthe—an anise-flavoured spirit derived from several plants, comprising 45–74% alcohol, also known as “The Green Fairy”), smoking, and oesophageal cancer [74]. Numerous longitudinal and case–control studies provide ample evidence linking such cancers to alcohol use [75].

Overall, regardless of the variety of beverages, ethanol intake accounts for an estimated 5% of total cancers, predominantly of the liver [76], upper respiratory tract, gastrointestinal tract, pancreas, breast, and lung [38,77]. Jayasekara et al. reported that their findings support dose-dependent associations linking chronic alcohol use with cancers such as breast, upper respiratory tract, digestive tract, and colorectal carcinoma [78]. Until now, many epidemiological studies, including several meta-analyses [79,80,81,82,83], strongly support the connection between alcohol use and the possibility of OC, yet it should be mentioned that certain studies have not found a significant link between alcohol and the incidence of oral tumours [84]; however, such studies are rare.

The above shows that alcohol is of great importance in the formation of cancer, including OC. This information has been known for a long time, but it still has not penetrated the general consciousness.

In 2012, 203.511 alcohol-related cases of the oral cavity, oropharynx, hypopharynx, and larynx cancers were reported (males: 179.559; females: 23.952) [85]. The percentage of HNC diagnoses linked to ethanol continues to increase. Certain research suggests that only medium to heavy alcohol use boosts the risk of HNC death [86]. In contrast, others indicate that even small levels of drinking correlate with a higher risk of death from these cancers [87]. According to Pelucchi et al., the incidence risk rate of OC rises by 20% with light alcohol consumption (<one drink per day) [88]. In their systematic review, Tramacere et al. drew attention to the detrimental impact of light alcohol use on the chance of developing oral and pharyngeal carcinoma [89]. According to Radoi et al., the population risk of OC linked to alcohol intake itself is below 18% [90]. Epidemiologic research shows that the risk associated with OC tends to rise if considered as an individual effect in people consuming >30 g of ethanol daily [91,92]. As claimed by Conway et al., a larger intake of more than three drinks daily over a short period (several years) is associated with a greater risk of oral cancer than a smaller intake over a long period (many years) [93]. Turati et al., in their systematic review of longitudinal and case-control studies from 1988 to 2009, found that the adjusted relative risk was 9.2 per >60 g (>four drinks) per day in Europe and 3.24 in the USA. Three studies found a significant dose-response correlation [81]. They did not find a clear association between the length of alcohol consumption and the risk of HNC in non-smokers [27,94,95,96]. According to Kawakita and Matsuo, alcohol is a recognised risk driver for head and neck cancer, together with its subtypes, even after tobacco smoking is corrected. The correlation is higher for oropharynx and hypopharynx cancers than for oral cavity and the larynx cancers. A larger ethanol intake in the short term was more harmful than a smaller one in the long time [27].

Ceasing the use of alcohol reduces the risk of HNC every year. After 20 years of abstinence, the risk becomes comparable to the risk of non-drinkers. The results showed that in upper aerodigestive tract (UADT) cancer survivors, drinking was linked to an over twofold increased risk of second UADT cancers versus not drinking [94].

Though alcohol use is a significant risk factor for head and neck cancer, only a few drinkers eventually develop HNC, which indicates that other conditions may influence the development of HNC [95]. Even though an association appears to exist between long-term alcohol consumption and the progression of OC, we have not yet entirely understood the exact role of alcohol in the pathogenesis of the disease. Not all OC sufferers use alcohol, and not all alcohol users develop OC [96].

## 5. Mechanism of Alcohol Toxicity

The ethanol molecule itself has not been proven to be genotoxic, mutagenic, or carcinogenic [97]. However, ethanol is metabolised into acetaldehyde (ACH) via alcohol dehydrogenase (ADH, EC 1.1.1.1); in turn, this is metabolised into the non-harmful molecule acetate via acetaldehyde dehydrogenase (ALDH, EC 1.2.1.10) which, after conversion into acetyl coenzyme A (Acetyl-CoA), enters a tricarboxylic acid cycle (the Krebs cycle or the TCA cycle). ACH causes DNA mutations and is accepted to have a major impact on carcinogenesis [97,98,99,100]. ACH fulfils four of the ten essential properties generally displayed by recognised human carcinogens—ACH is electrophilic and harmful to genetic material, interferes with DNA repair, and causes oxidative stress [101]. ACH’s in vitro and in vivo genotoxicity is generally known [97], and it has also been shown in other species [102].

Hence, the IARC has classified ACH from alcohol consumption as belonging to Group 1 carcinogens (“carcinogenic to humans”) [94,103,104]. The conclusions of the IARC refer to the ACH metabolically generated from ethanol and the “free” ACH found in alcoholic drinks [105]. ACH is also the most common carcinogen in cigarette smoke, which solubilises in saliva during smoking [106]. Exposure to both smoke and alcohol damages DNA and leads to incorrect DNA recovery [107]. Three factors in the mouth account for the accumulation of acetaldehyde: (1) ACH generated by the breakdown of ingested alcohol with oral mucosal ADH, (2) ACH from the breakdown of ingested alcohol with the oral microbiome, and (3) ACH from the alcoholic beverage itself.

Although alcohol metabolism predominantly occurs in the liver, the oral tissue layer and kidneys can also metabolise ethanol [108]. Cytoplasmic and mitochondrial ALDH break down ACH so efficiently that neither peripheral nor liver venous blood contain measurable ACH amounts after a dose of alcohol in persons with regular ALDH activity, and the amount of ACH is only slightly elevated (<10 mM) in those with ALDH deficiency [109,110,111]. Inversely, ACH concentrations were found in saliva after the consumption of alcohol at concentrations greater than what is needed to induce a mutagenic effect.

Typically, measurable amounts of ACH are not found in human saliva [112]. However, ethanol consumption (or tobacco smoking) causes an accumulation of ACH in the saliva which is dependent on the concentration, in seconds, that persists from 10 to 15 min following every sip of an alcoholic drink. Both ADH and ALDH dehydrogenases are expressed in the oral mucosa; however, ALDH is less active than ADH. Therefore, transformation into the relatively non-harmful acetate molecule is minimal in the oral cavity. This may lead to the build-up of mutagenic concentrations of ACH in saliva (compared to blood, its levels are 10 to 100 times higher) [113,114]. The locally generated ACH is mainly produced by microbes representing the normal gastrointestinal flora [9,115]. Bacteria and yeast in the mouth (e.g., *Candida albicans* and other Candida species) acquire ethanol from alcoholic and “non-alcoholic” drinks and foods to produce ACH. According to our findings, they also can convert sugars to ACH from food and drinks that do not contain alcohol at all [35]. It should be repeated that the ability of the mouth microbiota to eradicate ACH is equally low.

It must be added that during the treatment of patients with regular ALDH expression with 4-methylpyrazole (4-MP), an inhibitor of human ADH, no substantial alterations in ACH levels were observed in the saliva or blood. This proves that in people with normal ALDH and ADH activities, the generation of ACH in the saliva is mainly of microbial provenance [9,116]. Approximately 10- to 20-fold larger concentrations of 4-MP are required to achieve an inhibition of microbial ACH production of 40 to 50% in vitro than for an identical inhibition of ethanol elimination in humans [116]. Among people with an active ALDH2 enzyme, 10–15 mg/kg of 4-MP failed to lower ACH levels in saliva; however, it reduced the ethanol removal rate by 46%. Together, the obtained results imply that ACH exposure from ethanol can happen in the oral cavity separately from metabolism in the liver and that the oral ethanol metabolism may be viewed as a distinct risk factor for cancer.

So, why are alcohol and its metabolite ACH dangerous to living organisms? Alcohol has been suggested to increase epithelial cell vulnerability to carcinogens [117]. Still, various pathophysiological biomechanisms are also associated with the carcinogenic effects of ethanol, both direct and indirect [117]. A brief contact of the oral mucosa with ethanol increases the permeability of the epithelial barrier [9,61,118]. Alcohol also serves as a solvent for tobacco and facilitates tobacco-induced oral carcinogenesis [61,119]. Changes in membrane characteristics can increase the susceptibility of cells to carcinogens and bacterial infections, modify intracellular signal transmission, and eventually influence the immune response. ACH interferes with the creation and repair of DNA and attaches to proteins, causing changes in their structure and functioning. ACH attaches to DNA, creating stabilised DNA adducts capable of causing mutations. Reactive oxygen species (ROS), which are generated during oxidative ethanol metabolism, can upregulate vessel endothelial growth factor (VEGF), a tumour angiogenesis and cancer metastasis facilitator. Ethanol and ACH modify the transfer of methyl, triggering the hypomethylation of DNA. This process could change the production of oncogenes and cancer-suppressor genes. According to Wang et al., a chronic, low intake of alcohol entails global hypomethylation, while a higher intake of alcohol triggers gene-specific DNA hypermethylation, resulting in carcinogenic effects in both cases [120]. Ethanol can lower vitamin A acid levels thanks to the increased metabolism of the CYP2E1 system. Chronic alcohol use is linked to the degeneration of both the inborn and acquired immune systems [121]. The consequences include, among others, alcoholic liver damage. As a result, the organ is less capable of tackling potential carcinogens. Folic acid deficiency associated with ethanol is also related to cancer development. Long-term alcohol use, leading to parotid sialadenosis, may possibly result in a reduced salivary flow and reduced effectiveness in removing carcinogens from the oral cavity. [95,122,123,124,125,126]. Together, these processes are linked to the growth of cancer tumours, including OC, either autonomously or in combination. However, it should be noted that the innate susceptibility to the metabolisation of alcohol-derived carcinogens, especially ACH, can vary from person to person [127]. This may partly explain why some patients develop cancer at a younger age.

Yokoyama et al. intensively investigated a widespread ALDH gene polymorphism with a dormant mutated enzyme [128]. Inactive ALDH polymorphisms and reduced ACH clearance are strongly associated with the risk of cancer (oropharyngolaryngeal, esophageal, stomach, colon, and lung but not liver or other cancers). A single-point mutation of the ALDH2 gene lowers the activity of the main enzyme that metabolises ACH (ALDH2). Consequently, people with an ALDH2 deficiency after alcohol consumption are exposed to 2- to 3-fold concentrations of ACH in their saliva and 5- to 6-fold concentrations in their gastric juice compared to people who have an active ALDH2 enzyme [111,114,129,130]. In parallel with an increased local exposure to ADH, the risks of oral, pharyngeal, oesophageal, and stomach cancers among alcohol users with an ALDH2 deficiency are many times higher than in individuals with a functioning ALDH2 enzyme. The current epidemiological and biochemical evidence for people with an ALDH2 deficiency presents a rare and measurable model of the risk ACH presents to humans because it is not corrupted by interfering factors that hinder the majority of epidemiological research on cancer attributable to alcohol (ethanol particles are not carcinogens; ACH linked with alcohol drinking is carcinogenic (Group 1); salivary ACH levels are zero in the absence of ethanol or tobacco; the oral tissue layer is devoid of ALDH enzymes; the low or lack of ability of the oral microflora to eliminate ACH; the lack of disparity between ALDH2-deficient and ALDH2-positive persons when comparing the ability of the mouth microbiota to generate ACH from ethanol as well as the ability of the mouth microbiota to eliminate ACH) [130]). After drinking alcohol, local ACH exposure occurs immediately due to ACH’s immediate and mainly microbial formation [9,131]. Prolonged exposure represents ACH created from ethanol, which re-diffuses from the blood into the saliva within 30 min of the last sip of alcohol. Since the microbes of the mouth have a limited ability to eliminate ACH, as mentioned above, and as they cannot eliminate ACH microbially formed from ethanol, there are no indications of differences between ALDH2-deficient and ALDH2-positive persons. Thus, drinking causes the accumulation of mutagenic concentrations of ACH in the saliva of both ALDH2-positive and ALDH2-deficient persons insofar as alcohol remains in the human body [132].

## 6. The Role of the Microbiome

New academic studies concentrate on examining the function of the oral microbiome in the progress of OC and other cancers of the upper respiratory tract [60,114,133,134,135]. The oral microbiome hosts more than 750 common oral species. A healthy microbiome usually consists of Streptococci, Staphylococci, Neisseria species, and about 50 other aerobes [133]. The consensus is that a normal mouth microbiome consists mainly of aerobes, while the percentage of anaerobes increases with OC (and other pathological oral conditions).

The connection that exists between microbes and carcinoma is complex and influenced by a number of factors, such as the vulnerability and genetic makeup of the host, as well as environmental conditions, including the host’s dietary regime, oral hygiene, and tobacco and alcohol use [136,137]. Dysbiosis is usually given as a reason for oral carcinogenesis, as it promotes inflammation on one hand and ACH production on the other.

Changes in the microbiome’s composition can modify the microenvironment of the oral cavity, resulting in inflammation and malignancies [138]. The tumour microenvironment is strongly affected by cells connected to inflammation, which is an essential factor in carcinogenesis [82]. Many studies show that poor oral health fuels a persistent macrobiotic imbalance and is linked to dysplasia and the formation of cancer in the upper gastrointestinal passageway [139]. In contrast, the relationship of the oral microbiota with HNC was examined in a 4-year study by Hayes et al., and this study did not show a substantial risk related to particular types of bacteria or the makeup of the oral microbiome in subjects with this type of cancer [140]. Therefore, we can say that it is evident that the existence of localised infections may adversely affect the outcome of OC treatment, but it remains unclear whether the presence of dysbiosis is a cause or effect of the existence of the tumour. The main reason for this is that only a few studies have analysed the alteration of microbial populations in the initial phases of tumour formation [138].

Other authors indicate that the expression of ADH by some species of bacteria and yeasts is the leading cause of carcinogenesis [141]. As previously mentioned, the bacteria and yeasts present in the normal mouth microbiome play an instrumental part in the formation of topical ACH from the ethanol found in alcoholic (and also “non-alcoholic”) drinks or food. In a matter of seconds, ethanol consumption causes an accumulation of ACH in saliva which is dependent on the dose and is succeeded by a prolonged period of ethanol remaining in the human body [137,142]. Salaspuro reported that due to their ADH activity, a number of microbes that make up the normal oral microbiota can oxidise ethanol to ACH [143]. Peak salivary ACH levels following alcohol consumption may differ significantly among individual subjects, ranging from 18 to 260 µM [144]. The differences are caused by changes in the microbial composition of the oral cavity as well as the salivary ethanol concentration after alcohol consumption. In the oral cavity, the interindividual diversity of the microbial flora occurs primarily at the species or strain levels [137,145]. Homann et al. reported that salivary ACH formation from ethanol under in vitro conditions could be twice as high in those with poor oral health than in individuals with sound oral hygiene [113]. Rinsing the mouth with an antimicrobial rinse before consuming alcohol has been shown to reduce the number of microorganisms in the saliva and ACH rates by approximately 50%, proving the prominent role of the mouth’s bacterial microbiota in this process [146]. Homann et al. also highlighted the crucial role of oral microorganisms in the production of ACH as they showed a significant reduction in the ACH produced in vivo after the use of an antiseptic which was associated with limiting the number of bacteria in the saliva; the formation of ACH was halved after three days of using an antiseptic chlorhexidine mouthwash [144]. On the contrary, no association was established between the total number of bacteria in one’s saliva and the level of salivary ACH, demonstrating the possibility of no clear link between more significant numbers of bacteria and elevated levels of ACH. To the contrary, Yokoyama et al. found a relationship linking the number of bacteria and yeasts in the mouth and the formation of ACH in the saliva, which was demonstrated to decline following a three-week period of sobriety in male alcohol addicts in Japan [147].

Counts of bacteria and yeast may seem important, but to evaluate the individual risk of the presence of ACH in the oral cavity it is crucial to determine the makeup of the mouth microbiome. Therefore, when examining the microbial structure of the oral cavity, in addition to checking the number of bacteria, it is vital to study the microbial species to determine whether they are known producers of ACH. Yokoyama et al., after examining 100 saliva samples from healthy volunteers, demonstrated that oral microorganisms have different abilities to produce ACH from ethanol; the difference was up to 30 times [141], which at least partially accounts for a large variability of the amounts of ACH in the saliva of individuals following the consumption of alcohol.

It should be mentioned that not only does poor or good oral hygiene influence the composition of the microbiome but alcohol consumption is also known to modify it [9,137,147,148]. Regarding the effect of alcohol use on the population of oral cavity microbiota, a major study of Americans found that the microbial composition is altered, especially in heavy drinkers [149]. Notably, the number of Lactobacillus bacteria, which appear to be linked to anti-inflammatory and antioxidant properties, has dropped considerably. Smoking is also known to change the microbiome makeup and may increase the incidence of fungal infections caused by yeasts such as *Candida albicans* (*C. albicans*). Homann et al. reported that the microbiome of cigarette smokers has a greater vital capacity to generate acetaldehyde from alcohol both in vitro and in vivo [113].

Of the bacterial species that usually inhabit the oral cavity, the species Neisseria mucosa (Gram-negative aerobic bacteria, generally associated with good oral health) has an exceptionally elevated ADH activity and generates substantial levels of ACH when cultured in the presence of ethanol in vitro [60]. Muto and co-workers reported that Neisseria’s ability to produce ACH was more than 100 times more remarkable than other species tested [150]. Topical ACH generation by Neisseria mucosa in the mouth has been linked to the development of cancer. That said, a recent article noted an inverse correspondence between high levels of Neisseria and the ability of the mouth microflora to create ACH [141]. It needs to be added that ethanol consumption boosts the amount of Neisseria in the mouth.

Streptococci (Gram-positive aerobic bacteria) present in the healthy oral microbiota (for example, *S. salivarius*, *S. gordonii*, *S. intermedius*, and *S. mitis*) also exhibit a considerable enzymatic activity of ADH. Among 16 Streptococcus strains studied by Kurkivuori et al., *S. salivarius*, *S. intermedius*, and *S. mitis* showed significant ADH activity. Moritani et al. evaluated the production of ACH in vitro by 41 bacterial species from 16 genera chosen for their dominance and presence in the saliva of 166 healthy individuals. Among the examined species, all Neisseria, Rothia mucilaginosa, Streptococcus mitis, and Prevotella histicola were capable of generating ACH from alcohol in amounts over 50 µM [137,151].

The role of the yeasts inhabiting the oral cavity in ACH creation was first investigated in 1999. Of the yeasts capable of metabolising ethanol to ACH, positive evidence only exists for Candida species. Notably, both *C. albicans* and non-C. albicans (for example, Candida glabrata and Candida tropicalis) generate mutagenic quantities of ACH from alcohol (>100 μM) which can further contribute to epithelial dysplasia and oral cancer [60,137,152,153]. Yeast colonisation increased for the group with elevated salivary ACH production (78%) in comparison to the group with low (47%) salivary ACH production [154]. Tillonen et al. similarly concluded that heavy smoking or heavy drinking was associated with a greater yeast load [154]. Yeast overgrowth (especially Candida) has been reported in vivo in alcoholics [155]. Additionally, yeast species obtained from heavy alcohol users and cigarette smokers generated greater amounts of ACH compared to the control groups [156].

Long-term Candida infections are associated with mouth and oesophageal carcinogenesis in susceptible subjects [157]. Alnuaimi and colleagues examined the capacity of Candida isolates from people living with oral cancer and healthy individuals to generate ACH. The findings showed that Candida isolates generating larger quantities of ACH were more common in people with oral cancer than in healthy participants, additionally supporting the importance of Candida in ethanol-induced oral carcinogenesis, as well as the significance of identifying strains of microbiota for the assessment of oral ACH exposure [158,159]. Marttila et al. showed that under reduced oxygen pressure, which is common to many oral cavities, the levels of ACH generated in vitro from glucose by Candida could be orders of magnitude higher when compared to the levels generated when oxygen is present [160]. According to Makinen et al., there are findings that the appearance of yeast in the mouth is not statistically significant for the mortality of people with cancer [161]. Nevertheless, there was a substantial positive correspondence between OC and the occurrence and the extent of yeast colonisation in the oral cavity. Martilla et al. concluded that 68% of the cultures could generate carcinogenic amounts of ACH (>100 µM) under exposure to ethanol, and the levels of ACH produced in smokers were considerably increased compared to non-smokers [162].

We must add that although *C. albicans* is a commensal component of the oral mycobiome, it may as well be an opportunistic pathogen and transform from a harmless commensal into a pathogenic organism causing oral mucosal infection. Its number increases under dysbiotic conditions. It must be added that Candida can generate cancerous substances, e.g., nitrosamines or N-nitrosobenzylmethylamine. Carcinogenic compounds are able to attach to DNA to build adducts linked to DNA alterations and cancer development [163]. *C. albicans* can also influence the local specific microbial flora inside the biofilm and boost the number of anaerobic bacteria [164]. Such a process could result in a higher local build-up of ACH from microbial fermentation. Oral microbial strains with ADH activity are listed in Table 1.

## 7. Discussion and Conclusions

Auto-brewery syndrome (ABS) is an uncommon condition that affects men, women, and children in which ethanol is formed in the alimentary passageway through endogenous fermentation by selected yeast or, in some cases, by selected bacteria. Sufferers of auto-brewery syndrome commonly exhibit many symptoms of ethanol intoxication while simultaneously denying alcohol intake and also reporting a carbohydrate-rich diet. Several strains of bacteria and yeast which may be related to ABS have been described in medical publications. Among these are fermenting yeasts, e.g., Saccharomyces cerevisiae and Saccharomyces boulardii, various strains of Candida, including Candida glabrata, *C. albicans*, *C. kefyr*, and *C. parapsilosis*, and bacteria such as Klebsiella spp. and Enterococcus faecium. Yeasts that include Saccharomyces cerevisiae and *Candida albicans* shift to an oxygen-free metabolism involving the fermentation of alcohol with limited access to oxygen. Glycolysis is the initial fermentation step. The subsequent fermentation stage entails the decarboxylation of pyruvate into ACH, catalysed by pyruvate decarboxylase. In the concluding step, ACH is converted into ethanol by acetaldehyde-alcohol dehydrogenase. We could only find a single published article that described alcohol production due to oral candidiasis [165]. However, this experiment did not eliminate the probability of ethanol production in the gastrointestinal tract.

In contrast, our studies [35] confirmed that alcohol could be formed in the oral cavity (oral ABS). We had a case in which it caused a legal problem; however, in our research, it turned out that this alcohol disappeared in a few minutes and had low levels. Thus, someone must be extremely unlucky to have a legal problem for this reason (such as our individual). Still, the literature shows that the ACH lingers longer and will no longer be detected by police analysers, yet it will be a carcinogen.

As described above, all studies highlighted the impact of ACH on the development of OC, which can arise through various mechanisms: (1) ACH generated by the breakdown of ingested alcohol by oral mucosal ADH, (2) ACH from the breakdown of ingested alcohol by the oral microbiome, and (3) ACH from the alcoholic beverage itself. We add one more possibility: dysbiosis, which can be dangerous because it leads to the formation of ACH not only from alcohol consumed but also from ordinary food that does not contain alcohol (Figure 1). Due to the constantly increasing number of OC cases, this can be an important problem. Publications indicate that people with OC often deny drinking alcohol or underestimate the amount of alcohol they drink. Perhaps in some of these cases, OC is the result of the fourth mechanism, and the patients do not lie to their doctors. It would be worth considering the introduction of tests that could show whether a person has a daily appearance of ACH and therefore alcohol in the mouth and thus in the body. Of course, if doctors suspect oral ABS, they could use a breath analyser to test the patient after the patient consumes a glucose-containing meal; however, it is also worth considering a test that could demonstrate whether this condition is permanent. Additionally, hair tests for biomarkers such as ethyl glucuronide and the ethyl esters of fatty acids, which are present in the blood for a short time but remain in the hair, could help. After identifying a patient with oral ABS, causal treatment could be implemented first, i.e., antifungal drugs in the case of fungal colonisation and antibiotics in accordance with the antibiogram in the presence of alcohol-producing bacteria. The patient should be monitored and periodically microbiologically swabbed thereafter to determine if the problem has recurred. Such individuals could also consider using mouthwash, preferably without alcohol, of course.

The relationship linking the microbiome of the oral cavity with alcohol metabolism in the early stages of oral cancer is multifaceted. Assessing the exposure to alcohol-derived ACH in the mouth from yeast metabolism is problematic, considering the variety of factors that potentially contribute to oral cancers. As the reports in this field are new, we cannot yet assess the scale of the phenomenon. However, many case reports indicate that the problem is more than extremely rare. A high level of alcohol consumption, smoking tobacco, and dental health can alter the composition of the mouth microbiome, thereby indirectly affecting the salivatory microbial production of ACH to mutagenic levels. Together, these factors account for different vulnerability patterns and should be addressed in the evaluation of oral exposure to microbial ACH. More research is also needed to determine other species and strains of bacteria/yeasts that may contribute to the production of ACH from food and drinks in the oral cavity.

We must emphasise that no evidence exists that ethanol-derived ACH, which is sometimes present in low concentrations in soft drinks or foods, is not as carcinogenic to the upper digestive tract tissue lining as metabolically derived ACH from beverages containing alcohol. The same is true for the ACH present at mutagenic levels in fermented food, as well as ACH as a natural or added agent. For centuries, microbiological fermentation has been used for food conservation. While lactic acid is the most common output of alimentary fermentation processes, ethanol and ACH can also be produced in a predominantly oxygen-free environment. The levels of ethanol range between 0.5 and 2.5% or above. Such foods are homemade honey mead and beer, kephir, mursik milk, soy sauce, kimchi, pickles, and vinegar [166], which are popular in Eastern European and Asian countries. Unfortunately, when making decisions about which products qualify as alcoholic or non-alcoholic, people use the prediction that alcohol undergoes complete metabolism in the liver. However, this supposition is not true for ACH, which is produced in mutagenic amounts in the oral cavity even in the presence of low ethanol levels [144].

Therefore, we bring dysbiosis and the production of ACH from typical food and drinks (not containing alcohol) as a possible new contributor to the development of cancer; this is our hypothesis for further investigation, for instance, outlining the nuanced interaction between non-alcoholic drinks and food that can affect oral microbes in the natural history of the cancerous oral lesion.

## Figures and Tables

**Figure 1 biomolecules-13-00815-f001:**
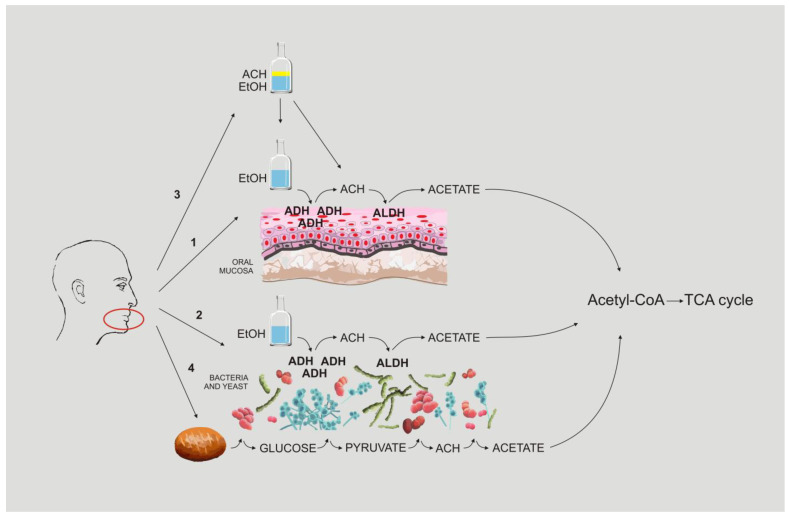
Pathways of oral acetaldehyde formation.

**Table 1 biomolecules-13-00815-t001:** Oral microbial strains with ADH activity.

Type	Species
Bacteria	*Neisseria mucosa*
	*Neisseria sicca*
	*Neisseria subflava*
	*Streptococcus salivarius*
	*Streptococcus intermedius*
	*Streptococcus gordonii*
	*Streptococcus mitis*
	*Enterococcus faecium*
	*Klebsiella pneumoniae*
	*Rothia mucilaginosa*
	*Prevotella histicola*
Yeasts	*Candida albicans*
	*Candida glabrata*
	*Candida kefyr*
	*Candida parapsilosis*
	*Saccharomyces cerevisiae*
	*Saccharomyces boulardii*

## Data Availability

Not applicable.

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
