# Peer review of "The Influence of the Oral Microbiome on Oral Cancer: A Literature Review and a New Approach"

_biomolecules, 2023, doi:10.3390/biom13050815_

Round 1
Reviewer 1 Report
In this meta-analytic review, the manuscript projected that oral microbes are responsible for carcinogen aldehyde production in patients having a history of alcohol inebriation.
this manuscript is well-written with appropriate references. There are some questions that may be required to be addressed:
1. author provides the list/table of what types of oral bacteria/candida albiacans strains are responsible for the onset or help in oral cancer development.
2. define the dysbacteriosis
Author Response
In this meta-analytic review, the manuscript projected that oral microbes are responsible for carcinogen aldehyde production in patients having a history of alcohol inebriation. This manuscript is well-written with appropriate references. There are some questions that may be required to be addressed:
Thanks for your kind review. We will respond to your comments below:
1. author provides the list/table of what types of oral bacteria/candida albiacans strains are responsible for the onset or help in oral cancer development.
We agree. The appropriate table is added.
2. define the dysbacteriosis
We agree. It is defined.
Reviewer 2 Report
Study well written and very interesting for conception and idea.
Needs some modifications and revision:
- line 25: Cancer cases, which ones? global? clarify
- line 28-33: from Breast [...] women (6.9%).delete, not needed.
- line 37-44 : Worldwide [...] factors. put paragraph before line 36-37 (Head [...] glands)
- line 54-80 : summarize
- line 109 : exclusion criteria. clarify
- line 125: (especially HPV 16, for 125 oropharyngeal carcinoma), indicate list of malignant histotypes in the literature.
- line 159-181 summarize. remove 165-166 : In addition, when consumed during pregnancy, it can cause severe 165 damage to fetuses and newborns [69].
- 237: remove brackets, add bibliography to line 237-238
- 268-269: bibliography needed
- 340: bibliography needed
- paragraph 5 (Mechanism of alcohol toxicity) and 6 (Role of microbiome) are interesting but too long: summarize and insert some concepts later in the discussion.
- Figure 1: compliments very clear and well structured.
- line 526 add POSSIBLE new factor for the development of cancer.
Thank you
Author Response
Study well written and very interesting for conception and idea. Needs some modifications and revision:
Thanks for your kind review. We will respond to your comments below:
- line 25: Cancer cases, which ones? global? clarify
We agree. Corrected.
- line 28-33: from Breast [...] women (6.9%).delete, not needed.
We agree. Deleted.
- line 37-44: Worldwide [...] factors. put paragraph before line 36-37 (Head [...] glands)
We agree. Moved.
- line 54-80: summarize
We agree. It is done.
- line 109: exclusion criteria. clarify
We agree. Clarified.
- line 125: (especially HPV 16, for 125 oropharyngeal carcinoma), indicate list of malignant histotypes in the literature.
We agree. List added.
- line 159-181 summarize. remove 165-166 : In addition, when consumed during pregnancy, it can cause severe 165 damage to fetuses and newborns [69].
We agree. It is done.
- 237: remove brackets, add bibliography to line 237-238
We agree. It is done.
- 268-269: bibliography needed
We agree. Added.
- 340: bibliography needed
We agree. Added.
- paragraph 5 (Mechanism of alcohol toxicity) and 6 (Role of microbiome) are interesting but too long: summarize and insert some concepts later in the discussion.
We do not fully agree. These issues are very important and it is difficult to shorten them, but we did our best.
- Figure 1: compliments very clear and well structured.
Thank you very much!
- line 526 add POSSIBLE new factor for the development of cancer.
We agree. Added.
Reviewer 3 Report
Thank you for writing this review, which comprehensive described the literature in relation to this connection.
- The first line of the abstract is a little grandiose, and should appear later in the introduction
- The introduction should include details of GBD oral cancer data in addition to GLOBOCAN
- Please not that reference 62 has been superseded (https://www.thelancet.com/article/S0140-6736(22)00847-9/fulltext)
Author Response
Thank you for writing this review, which comprehensive described the literature in relation to this connection.
Thanks for your kind review. We will respond to your comments below:
- The first line of the abstract is a little grandiose, and should appear later in the introduction
You are right. We changed it.
- The introduction should include details of GBD oral cancer data in addition to GLOBOCAN
We agree. Included.
- Please not that reference 62 has been superseded (https://www.thelancet.com/article/S0140-6736(22)00847-9/fulltext)
You are right. Changed.
Reviewer 4 Report
“The influence of the oral microbiome on oral cancer: a literature review and a new approach”, by Smedra and Berent is an interesting review on the effects of ethanol and acetaldehyde production in the oral cavity and its relationship with the low level of ALDH n the oral cavity. The title is misleading as the focus seems to be almost exclusively on ethanol and ADH and OSCC and oral pharyngeal OSCC and not other causes of OSCC and oral pharyngeal SCC. The review is well written but could stand some shortening as there are some consecutive sentences that are redundant. The authors do a good job of citing what has been done in the field and describing it. There seems to be a shortage of references to rodent studies, I am not sure how many have been done on ethanol effects on oral squamous cell carcinoma.
Ethanol promotes chemically induced oral cancer in mice through activation of the 5-lipoxygenase pathway of arachidonic acid metabolism
Yizhu Guo 1 , Xin Wang, Xinyan Zhang, Zheng Sun, Xiaoxin Chen, Cancer Prev. Res. Is one example.
Some of the writing is a little strange in the discussion. It would be nice if they supplied evidence that microbiome based ethanol production in the oral cavity being more than extremely rare.
Details:
Primary therapy is surgery, whereas radiation therapy 90
and chemoradiotherapy are complementary therapies to shrink tumours before surgery. 91
Sometimes they are used without surgery.
There was no clear association between the duration of alcohol use and the risk of head and neck cancer in non-smokers. 200.
Please supply references.
Yokoyama and colleagues in Japan intensively studied a common polymorphism in 296
the ALDH gene, where the mutated enzyme is inactive [123]. Inactive ALDH polymorphisms and reduced ACH clearance are strongly associated with cancer risk.
What type of cancers? Most should not be oral, as state elsewhere host ALDH has minimal activity in the mouth
It should be mentioned that not only bad or good oral hygiene influences the com- 402
position of the microbiome but also alcohol consumption is known to change it [8, 133, 143]. Regarding the impact of alcohol consumption on the oral cavity microbial population, a large study conducted on the American people has shown that the microbial composition is altered, especially in heavy drinkers [145]. In particular, there is a significant decrease in Lactobacilli, whose presence is associated with anti-inflammatory and antioxidant effects.
Actually these articles for the most part note an association of alcohol consumption and differences in the oral microbiome not a change in the microbiome. The exception is the Yokoyama study which was not well controlled likely due to Human Subejct Research restrictions.
In particular, both C. albicans and non-C. albicans (for example, Candida 433
glabrata and Candida tropicalis) contribute to epithelial dysplasia and oral carcinogenesis by producing mutagenic amounts of ACH from ethanol (>100 μM) [57, 133, 148, 149].
I was able to find evidence that ethanol consumption produces DNA adducts in oral mucosa in the citations but could not find evidence that Candida and ethanol exposure caused dysplasia. While that may be the case, the evidence is indirect and the wording of the statement needs to reflect that.
Author Response
"The influence of the oral microbiome on oral cancer: a literature review and a new approach”, by Smedra and Berent is an interesting review on the effects of ethanol and acetaldehyde production in the oral cavity and its relationship with the low level of ALDH n the oral cavity. The title is misleading as the focus seems to be almost exclusively on ethanol and ADH and OSCC and oral pharyngeal OSCC and not other causes of OSCC and oral pharyngeal SCC. The review is well written but could stand some shortening as there are some consecutive sentences that are redundant. The authors do a good job of citing what has been done in the field and describing it. There seems to be a shortage of references to rodent studies, I am not sure how many have been done on ethanol effects on oral squamous cell carcinoma. Ethanol promotes chemically induced oral cancer in mice through activation of the 5-lipoxygenase pathway of arachidonic acid metabolism Yizhu Guo 1 , Xin Wang, Xinyan Zhang, Zheng Sun, Xiaoxin Chen, Cancer Prev. Res. Is one example.
Thank you for your kind review. The title may be a bit misleading, but already in the content of the abstract we clearly explain what the work is about. Therefore, we suggest not to change it. We will respond to your comments below:
Some of the writing is a little strange in the discussion. It would be nice if they supplied evidence that microbiome based ethanol production in the oral cavity being more than extremely rare.
We agree. We changed it.
Details:
Primary therapy is surgery, whereas radiation therapy 90 and chemoradiotherapy are complementary therapies to shrink tumours before surgery. Sometimes they are used without surgery.
We agree. Changed and added.
There was no clear association between the duration of alcohol use and the risk of head and neck cancer in non-smokers. 200. Please supply references.
We agree. Supplied
Yokoyama and colleagues in Japan intensively studied a common polymorphism in 296 the ALDH gene, where the mutated enzyme is inactive [123]. Inactive ALDH polymorphisms and reduced ACH clearance are strongly associated with cancer risk. What type of cancers? Most should not be oral, as state elsewhere host ALDH has minimal activity in the mouth
We agree. Added.
It should be mentioned that not only bad or good oral hygiene influences the com- 402 position of the microbiome but also alcohol consumption is known to change it [8, 133, 143]. Regarding the impact of alcohol consumption on the oral cavity microbial population, a large study conducted on the American people has shown that the microbial composition is altered, especially in heavy drinkers [145]. In particular, there is a significant decrease in Lactobacilli, whose presence is associated with anti-inflammatory and antioxidant effects.
Actually these articles for the most part note an association of alcohol consumption and differences in the oral microbiome not a change in the microbiome. The exception is the Yokoyama study which was not well controlled likely due to Human Subejct Research restrictions.
We agree. Modified.
In particular, both C. albicans and non-C. albicans (for example, Candida 433
glabrata and Candida tropicalis) contribute to epithelial dysplasia and oral carcinogenesis by producing mutagenic amounts of ACH from ethanol (>100 μM) [57, 133, 148, 149]. I was able to find evidence that ethanol consumption produces DNA adducts in oral mucosa in the citations but could not find evidence that Candida and ethanol exposure caused dysplasia. While that may be the case, the evidence is indirect and the wording of the statement needs to reflect that.
We agree. Changed.
Round 2
Reviewer 4 Report
The influence of the oral microbiome on oral cancer: a literature review and a new approach, manuscript by Smedra and Berent has been improved. The authors have done a great job of explaining the concepts.
I still have an issue with the paragraph that starts:
Yokoyama et al. intensively investigated a widespread ALDH gene polymorphism 313 with a dormant mutated enzyme [128]
This paragraph in summary says that ALDH inactivating mutation or variants results in reduced levels of ACH clearance. This is associated with increased oral cancer rates and other tissues. They then explain that some of ACH in the mouth comes from blood.
The conclude the paragraph by saying:
Since microbes of the 335 mouth have a limited ability to eliminate ACH, as mentioned above, and they cannot 336 eliminate microbially formed ACH from ethanol, there is no indication of differences be- 337 tween ALDH2-deficient and ALDH2-positive persons. Thus, drinking causes the accumu- 338 lation of mutagenic ACH concentrations in the saliva of both ALDH2-positive and 339 ALDH2-deficient persons insofar as alcohol remains in the human body.
Line 328 the oral tissue layer is devoid of ALDH enzymes;
I am sure everything they wrote is true and the individual statements makes sense if their model is that overall ACH levels in the body have an effect on that in oral tissue in order for the oral cancer rates to be higher with the ALDH2 mutation. However, they certainly could have done a better job of explaining things. The authors may want to think about making things a little clearer but I leave it up to them.